# Overview of Recent Advances in Nano-Based Ocular Drug Delivery

**DOI:** 10.3390/ijms242015352

**Published:** 2023-10-19

**Authors:** Li-Ching Liu, Yi-Hao Chen, Da-Wen Lu

**Affiliations:** Department of Ophthalmology, Tri-Service General Hospital, National Defense Medical Center, Taipei 11490, Taiwan; ophelia30330@gmail.com (L.-C.L.); keanechen18@gmail.com (Y.-H.C.)

**Keywords:** ocular barriers, ocular drug delivery, nanocarriers, nano-based drug delivery system, ocular bioavailability

## Abstract

Ocular diseases profoundly impact patients’ vision and overall quality of life globally. However, effective ocular drug delivery presents formidable challenges within clinical pharmacology and biomaterial science, primarily due to the intricate anatomical and physiological barriers unique to the eye. In this comprehensive review, we aim to shed light on the anatomical and physiological features of the eye, emphasizing the natural barriers it presents to drug administration. Our goal is to provide a thorough overview of various characteristics inherent to each nano-based drug delivery system. These encompass nanomicelles, nanoparticles, nanosuspensions, nanoemulsions, microemulsions, nanofibers, dendrimers, liposomes, niosomes, nanowafers, contact lenses, hydrogels, microneedles, and innovative gene therapy approaches employing nano-based ocular delivery techniques. We delve into the biology and methodology of these systems, introducing their clinical applications over the past decade. Furthermore, we discuss the advantages and challenges illuminated by recent studies. While nano-based drug delivery systems for ophthalmic formulations are gaining increasing attention, further research is imperative to address potential safety and toxicity concerns.

## 1. Introduction

Ocular diseases profoundly impact vision and quality of life with over 2.2 billion people worldwide experiencing vision impairment according to the World Health Organization (WHO) [1]. The human eye, a complex and delicate organ, is shielded by numerous barriers to protect the visual axis from infections and inflammation [2]. The ocular structure could be broadly divided into the anterior segment and posterior segment. The former contains the cornea, conjunctiva, iris, ciliary body, and lens, while the latter includes the vitreous humor, retina, choroid, sclera, and optic nerve [3]. The unique structures of each layer create significant obstacles for ocular drug administration. The dynamic barriers such as choroidal and conjunctival blood flow, lymphatic clearance, tear dilution, and efflux pumps also interfere with precise dosage maintenance [4]. The conventional drug administration route includes topical administration, systemic administration, intraocular, and periocular administration [5,6]. Topical eye drops and ointments are the most common and convenient drug delivery methods. However, their effectiveness is limited by low drug penetration into the eye, reducing their therapeutic impact [7]. Therefore, a targeted drug delivery system (DDS) to overcome both anatomical barriers and physiological barriers in ocular tissue is always a major challenge for pharmacologists and researchers.

In the last two decades, ophthalmic research has rapidly advanced to develop secure, patient-oriented formulations, delivery methods, and devices. These aim to address previous challenges and ensure consistent drug levels in ocular tissues. Nanocarrier-based therapeutic delivery systems have proven to be a promising option to enhance drug retention and permeation and prolong drug release in ocular tissue [8]. Diverse nanocarrier types, such as nanodispersion systems, nanomicelles, solid lipid nanoparticles (SLN), nanostructured lipid carriers (NLC), polymeric nanoparticles, liposomes, niosomes, and dendrimers, have been extensively studied to enhance drug penetration and facilitate precise, targeted delivery to different parts of the eye [9]. Nonetheless, the process of transitioning these nanotechnology-driven DDSs from experimental stages to practical clinical applications faces hurdles related to upscaling production and ensuring consistent quality control [10].

In this review, we elucidate ocular anatomical and physiological features, emphasizing natural barriers. We explore limitations in conventional formulations and alternative drug delivery methods. Recognizing deficiencies in existing treatments, we spotlight advanced nanocarriers’ success against ocular diseases. Our review offers a comprehensive overview of various nanocarrier-based therapeutic systems, encompassing nanomicelles, nanoparticles, nanosuspensions, nanoemulsions, microemulsions, nanofibers, dendrimers, liposomes, niosomes, nanowafers, contact lenses, hydrogels, microneedles, and nano-based gene therapy. We highlight their unique traits and present current clinical applications for anterior and posterior segment ocular diseases, summarizing benefits and challenges.

## 2. Anatomy and Barriers of Ocular Drug Delivery

The normal human eye measures approximately 22 to 27 mm axially and 69 to 85 mm in circumference [11]. It can be divided into two segments: anterior and posterior, separated by the ciliary body and lens. The anterior segment comprises the cornea, conjunctiva, iris, ciliary body, and lens, while the posterior segment contains the vitreous humor, retina, choroid, sclera, and optic nerve. The eye’s intricate anatomy and protective barriers pose challenges for drug administration (Figure 1). It shares characteristics with immune-privileged organs like the brain, isolating it from the body’s circulation with a blood–retinal barrier, making systemic therapy difficult, particularly for posterior segment disorders.

### 2.1. Barriers of the Anterior Segment

#### 2.1.1. Tear Film Barrier

The tear film on the ocular surface forms an initial barrier, impeding drug delivery, while drainage through the nasolacrimal system can dilute and remove drugs, affecting their efficacy. This tear film, around 3 μm thick and 3 μL in volume, comprises three layers: an outer lipid layer, a middle aqueous layer, and an inner mucous layer [12]. The outer lipid layer prevents water evaporation but also hinders drug absorption into the cornea and sclera [13]. Meanwhile, the mucous layer in the tear film acts protectively, forming a hydrophilic barrier that efficiently removes debris and pathogens. 

The lacrimal turnover rate is approximately 1~3 μL/min, causing drug loss from the ocular surface to be 500 to 700 times greater than the drug absorption rate into the anterior chamber. Irritant drugs, certain excipients, and pH deviations can trigger lacrimation, increasing tear production to about 300 mL per minute [14]. This rapid increase leads to immediate drainage through the nasolacrimal duct, causing over 85% of the administered drug dose to be lost before reaching the corneal surface. The retained drug may also undergo further dilution due to rapid tear turnover, reducing the concentration gradient and diffusion rate. This results in low bioavailability of intraocular drugs within the aqueous humor, leading to poor drug bioavailability with topical delivery, typically ranging from 0.1% to 5% [15].

#### 2.1.2. Cornea and Conjunctival Barrier

The cornea, the outermost transparent avascular layer of the eye, has essential refractive and barrier functions. It consists of three cell layers: the lipophilic epithelium, the hydrophilic stroma, and the lipophilic endothelium, along with two interfaces: The Bowman layer and Descemet’s membrane. The corneal epithelium, comprising 5–7 lipid-rich cell layers with tight junctions and desmosomes, forms a robust barrier against drug penetration and microbial invasion [14,16]. The Bowman layer between the epithelium and stroma consists of acellular condensation of type I and type III collagen fibrils [17]. The Bowman layer allows drug and particle passage into the stroma, which makes up most of the cornea’s volume. It consists of hydrated type I collagen, providing structural support, optical clarity, and ocular immunity, facilitating the permeation and diffusion of hydrophilic drugs. Descemet’s membrane contains collagen type IV and VIII fibrils that provide support for the monolayer of corneal endothelial cells. Despite larger pore sizes in Descemet’s membrane reducing its barrier function, it can still filter macromolecules and particles that are directly administered into the stroma, protecting the endothelium. The corneal endothelium is a monolayer of cells that maintains stromal dehydration and allows the transportation of water and solute to the anterior chamber through both active (sodium–potassium ATPase pumps) and passive (endothelial intercellular tight junctions) mechanisms [16,18]. 

In contrast to the cornea, drug absorption through the conjunctiva is hindered by conjunctival capillaries and the lymphatic system, leading to drug leakage into the bloodstream and reduced bioavailability. Tight junctions in the conjunctival epithelium also impede the passive movement of hydrophilic molecules. The sclera, primarily composed of collagen fibers and proteoglycans, has a permeability similar to the corneal stroma. Recent studies suggest that drug permeation through the sclera inversely correlates with molecular size [19]. Linear dextrans exhibit lower permeability than globular proteins, and positively charged molecules have limited permeability due to interactions with the negatively charged proteoglycan matrix [4].

#### 2.1.3. Blood–Aqueous Barrier

The blood–aqueous barrier (BAB) is formed by tight junctions in the ciliary process’s non-pigmented epithelium, endothelial cells in the iris vasculature, and the inner wall endothelium of Schlemm’s canal. The tight junctions regulate paracellular transport, controlling the movement of ions and small substances between adjacent cells. The BAB is not completely impermeable; instead, it serves as a specialized gateway for controlled molecule movement [20].

### 2.2. Barriers of the Posterior Segment

#### 2.2.1. Vitreal Barrier 

The vitreous is a gel-like, transparent substance that fills the space between the lens and the retina. It mainly consists of water, collagen types II, IX, V/XI, hyaluronic acid, and other extracellular matrix components. Positively charged nanomaterials may interact with the negatively charged components of the vitreal network and thus block its diffusion ability, while negatively charged particles, based on the example of poly lactic-co-glycolic acid (PLGA) or human serum albumin, can distribute successfully across the vitreous humor [21]. The vitreous provides structural support to the eye, maintaining its shape against intraocular pressure. The vitreoretinal interface acts as a barrier, restricting substances from passing into the retinal layers [22]. This interface comprises three main components: (1) The cortical vitreous, a thin layer (100–300 μm) rich in collagen parallel to the inner limiting membrane (ILM). (2) The ILM, at the innermost boundary of the retina, is primarily composed of collagen type IV, laminin, and fibronectin, serving as a physical barrier. (3) Expanded Müller cell footplates, glial cells extending from the vitreous side to the outer nuclear layer of the retina.

#### 2.2.2. Blood–Retinal Barrier

The blood–ocular barrier (BOB) system includes two key barriers: the BAB and the blood–retinal barrier (BRB). The BRB is highly selective, controlling the passage of ions, proteins, and water to and from the retina. It comprises two parts: the outer BRB (oBRB), which includes the choroid, Bruch’s membrane (BM), and the retinal pigment epithelium (RPE), and the inner BRB (iBRB), formed by tight junctions among retinal capillary endothelial cells [23]. 

Starting from the outermost layer, the choroid includes the suprachoroid, large and medium blood vessel layers, and the choriocapillaris. The choriocapillaris play a role in nutrient supply and waste removal from the outer retinal layers. The Bruch’s membrane (BM), positioned between the choriocapillaris basement membrane and the RPE basement membrane, comprises outer and inner collagenous layers separated by a central elastic layer. BM allows size-selective passive diffusion but can block larger molecules. The RPE is a single layer of pigment-containing cells located beneath the neural retina layer. Its tight junctions maintain the integrity of the oBRB. For the iBRB, the retina vasculature penetrates at three main plexuses: the nerve fiber layer, inner plexiform layer, and outer plexiform layer. The iBRB mainly consists of the neurovascular unit, similar in structure and function to the blood–brain barrier, providing a barrier from systemic circulation. Molecule permeation is restricted based on size, charge, and lipophilicity. Small hydrophilic compounds can pass through junctions, while lipophilic molecules use the transcellular route [24]. 

The iBRB and oBRB have specific systems that allow substances to enter (influx transporters) or leave (efflux pumps) the retina. Developing drugs that mimic the substances recognized by influx transporters can help deliver drugs better into the retina. Additionally, designing drugs that are unrecognizable to the efflux pumps, or using inhibitors for these pumps, can help retain drugs in the desired location [21].

#### 2.2.3. Sclera and Bruch’s–Choroid Complex Barrier

The choroid serves as a densely vascularized barrier situated between the retinal pigment epithelium (RPE) and the sclera. With a thickness of approximately 200 μm, it is structured into five distinct layers: Bruch’s membrane, the choriocapillaris layer, two vascular layers, and the suprachoroidal layer [21,25]. The choroid acts as a barrier against hydrophilic compounds, while positively charged lipophilic drugs can bind with the tissue to create slow-release depots. Additionally, drugs’ molecule sizes impact their ability to diffuse into the posterior eye segment. Bruch’s membrane, approximately 2–4 μm thick, consists mainly of collagen and elastin fibers. The choriocapillaris layer comprises highly fenestrated capillaries with pores ranging from 6 to 12 nm, allowing the passage of larger molecules [26].

The sclera, the eye’s outer opaque layer, is primarily made up of collagen fibers, proteoglycans, and glycoproteins, with an average thickness of 0.5–1 mm. Drug permeability through the sclera is influenced by factors such as molecular weight, size, charge, and lipophilicity. For instance, hydrophilic compounds like methazolamide can penetrate the sclera. The proteoglycan matrix in the sclera carries a negative charge under normal pH conditions, aiding the passage of negatively charged solutes through this barrier [27].

## 3. The Conventional Routes for Ocular Drug Administration

The conventional ocular drug delivery methods include topical, regional, and systemic administrations. Topical delivery using solutions, gels, ointments, and suspensions is common and cost-effective, primarily targeting anterior segment eye diseases. However, less than 5% of the administered dose typically reaches the aqueous humor, as it can be washed off by mechanisms like lacrimation, tear dilution, and tear turnover [4,6,28]. In regional administration, the formulation is given intraocularly or periocularly to minimize systemic side effects while enhancing target tissue delivery [29]. Intracameral administration involves direct injection into the anterior chamber and is commonly used in cataract surgery and for treating anterior segment diseases. Another method, intravitreal injection (IVI), has gained popularity in the past two decades for treating retinal disorders such as age-related macular degeneration (AMD), diabetic macular edema (DME), retinal vein occlusion (RVO), and endophthalmitis. While IVI allows for direct drug delivery to the posterior segment, frequent eye punctures may increase the risk of complications like endophthalmitis, hemorrhage, retinal detachment, and patient discomfort [30]. Other intraocular injection deliveries include intrastromal, subconjunctival, subretinal, and subchoroidal. Periorbital injection methods include retrobulbar, peribulbar, posterior juxtascleral, and subtenon injection. This is widely used for local anesthesia in ocular surgery.

Systemic administration, via intravenous and oral routes, is used in various ophthalmic conditions like delivering antibiotics for endophthalmitis, carbonic anhydrase inhibitors for intraocular pressure, and methotrexate and corticosteroids for uveitis treatment [30]. However, as mentioned before, BAB and BRB restrict access to specific eye segments, reducing systemic administration’s bioavailability. This often necessitates higher dosages for clinical efficacy, potentially resulting in systemic side effects. Additionally, factors like plasma protein binding, lipophilicity, and distribution clearance can complicate predicting accurate therapeutic concentrations [31]. Table 1 summarizes various routes of ocular drug administration and their application in clinical treatment and describes their benefits and challenges in bioavailability. The pathways of drug metabolism are shown in Figure 2 [32].

## 4. Nanotechnology-Based Ocular Drug Delivery Systems

Ocular nanotechnology-based drug delivery is a specialized method for precisely administering therapeutic agents to the eye using nanoscale carriers or systems (Figure 3). These carriers transport drugs to the target site, improve solubility and stability, extend release, and minimize side effects. This approach has gained attention over the past two decades for overcoming ocular barriers and enhancing drug bioavailability [10,14,36]. Some nanocarriers have received FDA approval and are commercially available for use in the clinical treatment of ocular diseases (Table 2). We provide a comprehensive introduction and list some recent exploratory studies in each nanotechnology-based ocular delivery method, highlighting their clinical applications in various eye disorders to offer treatment insights and explore future developments in this field.

### 4.1. Nanomicelles

#### 4.1.1. Main Characteristics

Nanomicelles are self-assembling nanoscale (particle size usually within a range of 5 to 100 nm) colloidal dispersions with a hydrophobic core and hydrophilic shell. The micelles can be divided into three categories: polymers, surfactants, and multi-ion composite nanomicelles [37]. Amphiphilic molecules forming micelles exhibit properties influenced by factors like temperature, concentration, and the number of monomer molecules. In aqueous solutions, these molecules self-assemble into spherical or cylindrical nanoparticles, with their hydrophobic tails in the core and hydrophilic heads on the surface interacting with the aqueous environment. This structure allows for loading hydrophobic drugs into the core, improving solubility, stability, and targeted delivery to specific tissues [38]. Conversely, when amphiphilic molecules assemble in a nonpolar solvent, the hydrophobic part of the molecule faces outward while the hydrophilic portion is embedded inward, constituting so-called reverse nanomicelles, which can be used to load soluble drugs [39]. Stimuli-responsive nanomicelles release drugs in response to various cues, such as pH, temperature, ultrasound, light, and redox potential. These diverse nanomicelle structures enhance the solubility, stability, and delivery of hydrophobic compounds, generating significant interest in pharmaceutical and biomedical research.

Polymer nanomicelle carrier materials fall into two categories: natural and synthetic. Natural polymers like hyaluronic acid, albumin, and chitosan are widely available, biocompatible, and known for their non-toxic breakdown products in the human body, making them attractive for chemical modifications. Synthetic polymer nanomicelle carrier materials, such as polyethylene glycol (PEG), polyethylene oxide (PEO), polyacrylamide (PAM), polyvinylpyrrolidone (PVP), polyvinyl alcohol (PVA), and polyethyleneimine (PEI), comprise the hydrophilic sections of polymer nanomicelles. Hydrophobic components include biodegradable polyesters and amino acids like polylactide (PLLA), polyglycolide (PGA), polycaprolactone (PCL), polylactic glycolate (PLGA), polyaspartic acid (PAsp), polybenzylaspartic acid (PBLA), and polyglutamic acid (PGlu). Polymer nanomicelles can be categorized into three types based on the arrangement of amphiphilic molecules: amphiphilic block copolymers, graft polymer micelles, and amphiphilic random copolymers [40,41,42].

#### 4.1.2. Clinical Applications

In recent years, nanomicelles have made significant strides in ocular drug delivery. These advancements include approved treatments like rapamycin nanomicelle eye drops to prevent immune rejection [43], terbinafine hydrochloride nanomicelle formulations for fungal eye infections [44], and Cequa^®^ (a 0.09% cyclosporine eye solution) for dry eye syndrome management [45]. Table 3 summarizes recent in vitro and in vivo studies of nanomicelles in ophthalmic DDS applications. However, it is crucial to assess potential cytotoxicity, genotoxicity, and immune response before clinical use, especially considering the presence of multiple neuronal cells in the retina. Additionally, evaluating the ability of nanomicelles to cross the BRB and reach different retinal layers is essential to understand their potential toxicity. Thus, comprehensive toxicology testing is essential for assessing new drug carrier candidates in the future [46].

### 4.2. Nanoparticles

#### 4.2.1. Main Characteristics

Nanoparticles (NPs) are tiny particles ranging from 1 to 1000 nm and can be composed of various materials like metals (e.g., gold or silver), polymers, lipids, ceramics, or other substances. They come in different shapes, such as spheres, rods, tubes, or irregular forms, depending on their material and fabrication method. They may or may not have a core–shell structure, which is different from micelles. NPs can be divided into nanocapsules and nanospheres based on their morphological structure. Nanospheres have a solid polymeric structure, whereas nanocapsules consist of a thin polymeric envelope, approximately 5 nm thick, surrounding the oily core. NPs can encapsulate both hydrophobic and hydrophilic drugs, safeguarding them from degradation, and improving targeted delivery, efficient drug absorption, and controlled drug release, thereby enhancing bioavailability [59]. In ophthalmic DDSs, NPs are usually composed of lipids, such as fatty acids and triglycerides, or polymers, which could be further divided into natural and synthetic. Natural polymers including cellulose, sodium alginate, hyaluronic acid, albumin, gelatin, and chitosan have the advantages of biocompatibility and minimal toxicity. Different natural polymers possess unique characteristics. For example, chitosan, a cationic polysaccharide derived from chitin, exhibits favorable mucoadhesive and antimicrobial properties in ocular drug delivery systems. However, it is insoluble in both water and alkaline solutions. On the other hand, hyaluronic acid boasts a high hydration capacity but comes with a higher cost [60]. The synthetic polymers, such as polylactic acid (PLA), polymethacrylic acid (PMAA), polycyanoacrylate, Eudragit^®^ (RS100 and RL100), poly(ε-caprolactone) (PCL), poly(lactic acid-co-glycolide) (PLGA), and polyacrylamide, can be engineered to have versatile properties to improve biodegradability, biocompatibility, and controlled release manners which can meet specific therapeutic needs. However, they may also face some clinical concerns such as toxicity of monomers or byproducts and may provoke an immune response in vivo.

Solid lipid nanoparticles (SLNs) and nanostructured lipid carriers (NLCs) are both lipid-based nanoparticles used in drug delivery. The former is characterized by a solid matrix that enables controlled drug release, alongside improved stability and cost-effectiveness, while the latter is regarded as a second-generation lipid nanoparticle, consisting of both liquid and solid lipids without any crystalline structure [61].

NP size significantly affects drug loading and delivery in ocular drug delivery. Smaller NPs generally offer better stability and biodistribution. NPs in the range of 50–400 nm are preferred for ocular drug delivery, with sizes around 200 nm easily absorbed by the cornea and conjunctiva through topical administration. They exhibit improved mucoadhesion, better penetration through ocular barriers, and reduced ocular irritation [42,62]. The surface charge of NPs is essential; cationic NPs tend to stay longer on the ocular surface due to their interaction with negatively charged tissues, while anionic NPs have different characteristics [63].

#### 4.2.2. Clinical Applications

Recent advancements have made significant progress in using NPs for detecting ocular diseases. For example, Nguyen et al. developed ultrapure chain-like gold nanoparticles (CGNPs) conjugated with RGD peptides for multimodal photoacoustic microscopy (PAM) and optical coherence tomography (OCT) to visualize choroidal neovascularization (CNV) in a rabbit model. Intravenous administration of CGNP clusters-RGD bound to CNV resulted in significantly enhanced PAM and OCT signals, offering a more sensitive diagnostic tool for retinal neovascularization [64,65,66]. Gold nanoparticles (GNPs) have been employed in detecting and treating retinoblastoma. Moradi et al. showed that combining brachytherapy and hyperthermia with GNPs increased retinoblastoma necrosis and significantly reduced its size in rabbit eyes [67].

Nanoparticles in ocular treatment come in two types: polymer-based and lipid-based. Polymersomes are vesicular structures characterized by a bilayer composition consisting of hydrophilic and hydrophobic block copolymers, with a central core filled with an aqueous solution. The polymer NPs are characterized by their biodegradable properties, extended circulation in the bloodstream, improved drug-loading capacity, and easily modified surfaces through the attachment of ligands which lead to targeted delivery [68]. By adjusting the composition and processing conditions, researchers can tailor the size and shape of polymersomes to meet specific requirements. Common shapes include spherical, oblong, and tubular. Therefore, polymersomes are widely used in the biomedical field for both therapeutics and diagnosis. In ophthalmic applications, formulating pranoprofen with PLGA polymer has improved ophthalmic delivery and enhanced the drug’s local anti-inflammatory and analgesic effects [69]. Varshochian et al. developed albuminated-PLGA-NPs containing bevacizumab for intravitreal injection in rabbit models. This formulation showed a sustained release of bevacizumab for 2 months, maintaining vitreous concentrations above 500 ng/mL (the minimum concentration required to block VEGF-induced angiogenesis) for over 8 weeks. This suggests that these NPs can sustain drug release, reducing the need for frequent dosing [70]. Chitosan-based polymeric nanoparticles loaded with drugs such as cefuroxime, diclofenac, atorvastatin, or dexamethasone have shown improved ocular bioavailability [71,72,73]. Hyaluronic acid-based nanoparticles have demonstrated favorable attributes, including size, pH, osmolarity, and entrapment efficiency, leading to enhanced corneal permeation and increased accumulation of cyclosporine A. This holds promise for the treatment of dry eye disease compared to commercial emulsions in a rabbit model [74].

Polymeric NPs face challenges related to cytotoxicity and a lack of efficient large-scale production methods. In contrast, lipid NPs pose a lower risk of toxicity since they rely on biodegradable and non-toxic lipid components. Both SLNs and NLCs raise great interest in DDSs due to their outstanding biocompatibility, tolerability, and scaling-up capabilities. SLNs were first reported in the 1990s by Professor Müller and Professor Gasco. SLNs represent a mixture of solid-state lipids under ambient and physiological temperatures while NLCs are the second generation of lipid nanoparticles that include liquid lipids in their structure [75]. Lipid-based NPs loaded with brimonidine and latanoprost have been employed for the treatment of glaucoma [62,76]. Antimicrobials such as natamycin and ofloxacin coated with SLNs have been shown to increase transcorneal permeation, prolong drug release rate, and enhance antifungal activity without cytotoxic effects [77,78]. In retinal vascular diseases such as AMD and DME, SLN and NLC-modified medications also improved bioavailability in aqueous and vitreous humor in animal models [79,80]. These nanoparticles can interact with the ocular surface while safeguarding the drugs from metabolic degradation and extending their residence time on the pre-corneal surface [81].

### 4.3. Nanosuspension

#### 4.3.1. Main Characteristics

Nanosuspensions (NSs) consist of pure drug nanoparticles and stabilizers, typically with an average diameter below 1 μm (often in the range of 200–500 nm). They can be formulated in either aqueous or non-aqueous liquid phases, enhancing drug solubility in both aqueous and organic environments [82,83]. NSs are a versatile approach to improving the delivery of hydrophobic drugs and show promise in enhancing the performance of poorly water-soluble drugs, particularly those from natural sources [84]. NSs require careful formulation with stabilizing agents to maintain drug particle stability in the liquid medium. Various fabrication methods, including top-down techniques like wet milling, dry milling, high-pressure homogenization, and co-grinding, as well as bottom-up methods like anti-solvent precipitation, liquid emulsion, and sono-precipitation, are used. A combination of these approaches is often employed. NSs can be administered in liquid forms, or post-production processes like freeze drying, spray drying, or spray freezing can convert them into solid forms, allowing for diverse dosage forms like powders, pellets, tablets, capsules, films, or gels [85].

In ocular DDSs, nanosuspensions offer a method to deliver higher concentrations of poorly soluble drugs and extend residence time to the cul-de-sac [86]. NSs hold promise for improving ocular disease treatment through enhanced drug delivery efficiency, bioavailability, and patient comfort. However, challenges related to stability, particle size distribution control, and safety and toxicity concerns require rigorous testing and optimization to fully realize their potential and ensure their safety and efficacy in clinical applications.

#### 4.3.2. Clinical Applications

Extensive research is ongoing to explore the use of NSs in various drug delivery systems, including oral, ocular, brain-targeted, topical, buccal, nasal, and transdermal routes [86]. In ocular drug delivery, NSs are being investigated for treating diverse eye conditions, such as glaucoma, macular degeneration, diabetic retinopathy, and uveitis. Table 4 provides examples of recent exploratory studies of NSs for both anterior segment and posterior segment disease treatment.

### 4.4. Nanoemulsions and Microemulsions

#### 4.4.1. Main Characteristics

Nanoemulsions (NEs) are emulsions with sizes ranging from 20 to 500 nm, composed of two immiscible liquids (usually water and oil) stabilized by an amphiphilic surfactant [93]. They are transparent or translucent and thermodynamically unstable but kinetically stable systems. NEs are categorized based on the nature of the dispersed phase system, including oil-in-water (*o*/*w*) NEs, water-in-oil (*w*/*o*) NEs, and bi-continuous NEs, which feature both oil microdomains and water in the system. Among these, oil-in-water (*o*/*w*) NEs have gained prominence in ocular drug delivery due to their unique characteristics, such as easy dilution with tear fluid and the capacity to encapsulate lipophilic drugs in the oil phase.

Microemulsions (MEs), first introduced by Hoar and Schulman in the 1940s, are isotropic and thermodynamic stable dispersions made from water, oil, surfactants, and cosurfactants with small droplet sizes usually within the range of 5–200 nm [94]. While NEs and MEs share similar formulation components, including oil and water phases, surfactants, and potentially cosurfactants, their ratios differ. Generally, MEs require a higher surfactant-to-oil ratio compared to NEs [95]. In addition, NEs typically feature spherical-shaped particles due to the significant Laplace pressure, whereas MEs may display both spherical and non-spherical particles owing to their notably low interfacial tension. Ophthalmic NE formulations have demonstrated extended pre-corneal retention times, increased ability to penetrate ocular tissues, enhanced ocular drug bioavailability, and consistent drug levels in the eye. These benefits surpass those of previously reported gels or ointments. Cationic NEs extend drug dwelling time through electrostatic interactions with the anionic surface of the cornea’s mucin layer. This interaction enhances drug penetration through corneal tight junctions, ultimately increasing bioavailability [96,97].

#### 4.4.2. Clinical Applications

NEs have broad applications in biomedicine owing to their small droplet sizes, offering stability and rheology control. They are widely employed in pharmaceutical formulations for topical, ocular, and intravenous delivery. NEs also serve as templates for producing nanocrystals of hydrophobic pharmaceutical ingredients [98]. Due to their excellent permeability through the cornea and conjunctival barriers, NEs are extensively used in treating anterior segment disorders. Fardous et al. created a gel-in-water (G/W) NE by ultrasonication using beeswax as an organogelator aiming to generate a delivery system of hydrophobic drugs to the posterior ocular region. In vitro testing demonstrated the biocompatibility of the G/W NE. Meanwhile, in vivo, application as eye drops revealed no signs of ocular irritation [99]. Despite the growing literature and products in this field, there are still challenges with NEs, including the need for significant amounts of surfactants and concerns about their potential toxicity, limited capacity to solubilize high-melting-point substances, and susceptibility to environmental factors affecting stability. Therefore, careful component selection and safety evaluation are essential for pharmaceutical development.

### 4.5. Nanofibers

#### 4.5.1. Main Characteristics

Nanostructures are generally classified into four types: zero-dimensional (nanoparticles), one-dimensional (nanofibers, nanotubes, nanowires), two-dimensional (nanofilms), and three-dimensional (polycrystals). One-dimensional nanostructures are particularly intriguing due to their high surface-to-volume ratio, porous structure, mechanical strength, flexibility, and resemblance to the extracellular matrix [100,101]. Nanofibers, with diameters typically ranging from tens to hundreds of nanometers, are composed of various materials, including natural polymers (e.g., hyaluronic acid, chitosan, dextran, gelatin, collagen), synthetic polymers (e.g., PLA, PLGA, PCL), carbon, or ceramics. They exhibit a high aspect ratio and can be fabricated using various methods, categorized as either electrospinning or non-electrospinning. Electrospinning, in particular, has gained significant attention due to its ability to produce ultrafine fibers through the application of high voltage and low current, allowing for potential commercial-scale production [98,102].

#### 4.5.2. Clinical Applications

Nanofiber composites are widely used in various biomedical applications such as medical implants, wound dressing, tissue scaffolds, and drug delivery systems. Drugs ranging from antibiotics and anticancer agents to proteins, DNA, RNA, living cells, and various growth factors can be loaded into electrospun nanofibers [103]. The drug release rate of nanofibers can be influenced by their morphology, porosity, and composition. Nanofibers, with their large surface area, offer significant drug loading capacity and extended drug release profiles, reducing the need for frequent dosing. Table 5 summarizes some recent research on nanofiber-based ocular drug delivery systems for both anterior and posterior segment disorders.

### 4.6. Dendrimers

#### 4.6.1. Main Characteristics

Dendrimers are highly branched nanoscale polymer structures with a three-dimensional design and numerous functional groups on their surface, making them versatile and biocompatible for various applications [112]. Over a hundred distinct dendritic architectures exist, with polyamidoamine (PAMAM) and polypropyleneimine (PPI) dendrimers being widely recognized among them [113]. T Dendrimers can be synthesized to control their size, surface charge, peripheral functional groups, and solubility, making them highly adaptable for carrying various therapeutic drugs [114]. Dendrimer-drug conjugates have the potential to reduce systemic side effects, improve drug targeting, and prolong drug half-lives.

The cytotoxicity of dendrimers depends on factors like generation, surface group number, and terminal moiety type. Higher-generation dendrimers and those with positive surface charges tend to be more cytotoxic. PAMAM dendrimers at concentrations above 50 μg/mL may cause ocular functional damage [115]. Dendrimers modified with substances like polyethylene glycol (PEG), acetyl groups, carbohydrates, and other moieties have either minimal or no adverse effects on cells. Furthermore, some dendrimers possess inherent biological properties, such as anti-fungal, anti-bacterial, or cytotoxic effects on cancer cells while sparing normal cells. Therefore, the potential intrinsic cytotoxicity should be evaluated on a case-by-case basis [116].

#### 4.6.2. Clinical Applications

Recent studies have shown promising results using dendrimer-based carriers for ocular drug delivery via various administration methods, including topical, intravitreal, and subconjunctival routes [114]. For instance, DenTimol, a dendrimer-based polymeric timolol analog, demonstrated efficient corneal penetration and a significant intraocular pressure (IOP) reduction in adult male rats, achieving an average reduction of 7.3 mmHg (approximately 30% from baseline) within 30 min [117]. Soiberman et al. developed a subconjunctival injectable gel using G4-PAMAM dendrimer and hyaluronic acid loaded with dendrimer dexamethasone (D-Dex) conjugate in a rat corneal inflammation model, leading to a significant reduction in corneal neovascularization. This injectable D-Dex gel holds promise as a drug delivery platform for treating various inflammatory ocular surface disorders, including dry eye, autoimmune keratitis, and post-surgical complications, reducing the need for frequent steroid administration. Additionally, dendrimers are utilized as scaffolds in corneal tissue engineering [118]. Duan et al. reported that a dendrimer crosslinked collagen gels supported human corneal epithelial cell growth and adhesion, without cell toxicity [119].

In ocular posterior segment diseases, dendrimers are also a promising method for drug delivery. Kambhampati et al. reported that systemic hydroxyl-terminated polyamidoamine dendrimer-triamcinolone acetonide conjugate (D-TA) could be selectively taken up by the injured mi/ma and RPE causing choroidal pro-inflammatory cytokines and pro-angiogenic factor suppression by limiting macrophage infiltration, resulting in significant CNV suppression in a rat model. The dendrimer was also taken up by choroidal macrophages in human postmortem diabetic eyes [120]. Kannan et al. highlighted the progress in therapeutics using hydroxyl polyamidoamine dendrimers, which can target cells systemically without the need for specific ligands. This offers a novel treatment option for wet AMD [121]. Recently, genome-editing technology for genetic disorders or cancers has gained lots of interest. Inoue et al. developed a folate-modified polyamidoamine dendrimer (FP-CDE) as a potential carrier for TTR-CRISPR pDNA therapy in hereditary amyloidogenic transthyretin ocular amyloidosis [122].

### 4.7. Liposomes and Niosomes

#### 4.7.1. Main Characteristics

Liposomes and niosomes are vesicular drug delivery systems designed to encapsulate and deliver drugs in a controlled manner. They have similar structures but differ in composition. Liposomes are spherical vesicles made of phospholipid bilayers with a central water compartment diameter of 0.025 to 10 μm [123]. Liposomes can encapsulate hydrophilic and lipophilic drugs in their aqueous core or lipid bilayer, respectively. They can be modified to enhance specificity for specific cells or tissues by adjusting surface molecules, size, and lamellarity, and by adding targeting ligands or polymers [124,125]. Furthermore, liposomes are biocompatible and biodegradable, making them suitable for drug delivery applications and are widely used in pharmaceuticals, particularly for delivering chemotherapy drugs and vaccines, as well as in cosmetic products. Apart from the above benefits, liposomes present some disadvantages such as limited stability which may require a specific storage environment, limited capacity for loading certain hydrophobic compounds, and a short circulation half-life due to clearing by the immune system [126].

Niosomes are similar to liposomes in structure and function but are composed of non-ionic amphiphilic molecules in certain aqueous solutions by self-assembly technology instead of natural or synthetic phospholipids. The absence of phospholipids in niosomes makes them more stable than liposomes and less prone to oxidation or degradation. Niosomes can co-deliver both hydrophilic and lipophilic drugs in one vesicle. The formulation process is easier due to the good stability of the niosomes and the cost of preparation is also much cheaper than liposomes. Thus, niosomes are gaining importance in drug delivery systems [127]. However, niosomes can suffer from drug leakage over time, especially when exposed to physiological conditions. This leakage can affect the controlled release of drugs and reduce their therapeutic efficacy.

#### 4.7.2. Clinical Applications

The choice between liposomes and niosomes hinges on factors like the drug type and desired system properties. They are versatile and customizable to regulate drug release rates and target specific cells or tissues through surface modifications. Several liposomal products, including Visudyne^®^, an FDA-approved formulation of Verteporfin, are already available for clinical use in photodynamic therapy for the treatment of patients with predominantly classic subfoveal CNV due to AMD, pathologic myopia, or presumed ocular histoplasmosis [128]. Additional FDA-approved liposomal ophthalmic formulations include Lacrisek^®^, a liposome-based formulation of vitamin A palmitate and vitamin E, and Artelac Rebalance^®^, a vitamin B12 eye drop formulation with PEG and hyaluronic acid, used for the clinical treatment of dry eye disease [36,129]. There is growing preclinical research on liposome-based and noisome-based ophthalmic formulations. Table 6 offers some of the recent studies in this field.

### 4.8. Nanowafers

#### 4.8.1. Main Characteristics

A nanowafer is a small, transparent, circular disk containing nanoreservoirs filled with medication. These nanoreservoirs release the drug gradually, improving its absorption into the nearby eye tissue. The nanowafer dissolves after the desired drug release duration. PVA is a suitable non-immunostimulatory polymer for ocular nanowafer fabrication [140]. Nanowafer drug delivery offers sustained release, reducing the need for frequent dosing, minimizing systemic exposure, and reducing potential side effects. The use of biodegradable components enhances safety and minimizes long-term impacts. However, material choice in nanowafer formulations is critical for biocompatibility and to avoid toxicity. Dissolution rates can vary with formulation, affecting drug release kinetics and dosing regimens.

#### 4.8.2. Clinical Applications

Nanowafer technology finds applications in various fields, including drug delivery. A study by Yuan et al. showcased the effectiveness of an axitinib-loaded nanowafer in treating corneal neovascularization. A once-a-day administration of the nanowafer outperformed twice-daily eye drops in therapeutic efficacy without impacting wound healing or causing toxicity in experimental models [140]. Marcano et al. created a nanowafer for cysteamine delivery to treat corneal cystinosis, a condition typically managed with frequent and irritative eye drops. In vivo tests on cystinosis knockout mice showed that the nanowafer, containing 10 μg of cysteamine and administered once daily, was twice as effective as twice-daily eye drops containing 44 μg of cysteamine [141]. Coursey et al. also proposed that a dexamethasone (Dex)-loaded nanowafer could release the drug on the ocular surface for a longer duration than aqueous eye drops. They found out that administering the Dex-NW treatment once daily, every other day over five days, effectively restored a normal ocular surface and corneal barrier function. This regimen exhibited a level of effectiveness comparable to that of applying dexamethasone eye drops twice daily. Additionally, Dex-NW significantly reduced the expression of inflammatory cytokines (e.g., TNF-α and IFN-γ), chemokines (such as CXCL-10 and CCL-5), and MMP-3, which are typically elevated in response to dry eye conditions [142].

### 4.9. Contact Lenses

#### 4.9.1. Main Characteristics

Nanocarriers in contact lenses involve integrating nanoparticles or nanoscale materials into the contact lens structure. This technology aims to improve contact lens performance and comfort while facilitating controlled drug delivery directly to the ocular surface. This approach can extend drug retention time and enhance ocular bioavailability by over 50% [143,144]. There are mainly two types of contact lenses depending on the designed material: (1) soft contact lenses (SCL), composed of hydrogel or silicone hydrogel polymers, and (2) rigid gas-permeable contact lenses (RGP) [145]. The parameters of soft contact lenses include transparency, oxygen permeability, and glass transition temperature. SCL surface modifying methods include dip-coating (soaking); diffusion barrier insertion (Vitamin E); incorporation of functional monomers, ligands, and a polymeric matrix; molecular imprinting, incorporation of colloidal, drug-loaded nanoparticles or other colloidal nanostructured systems; and surface coating by multilayer film deposition of colloidal nanoparticles or ligands [146,147].

Contact lenses as drug delivery systems have numerous advantages for ocular drug delivery, including sustained drug release, targeted delivery, ease of wear for improved patient compliance, reduced tear dilution, protection of sensitive drugs, and potential for combination therapy. Despite these advantages, contact lens drug delivery systems also face challenges such as controlling drug release rates precisely, potential infection risk from improper handling, and the need for material and design optimization [147].

#### 4.9.2. Clinical Applications

Recent applications of nanoparticle medicines in soft contact lenses include antibiotics, antihistamines, immunosuppressants, corticosteroids, and glaucoma drugs, with a focus on anterior segment ocular disorders. While contact lenses offer precise drug delivery control, they have limitations in terms of a short drug release duration, storage stability, and drug capacity [148]. Research is growing in the field of ophthalmic formulation techniques and lens material design to improve bioavailability in clinical applications. Table 7 summarizes recent preclinical research on soft contact lenses for various ocular diseases.

### 4.10. Hydrogels

#### 4.10.1. Main Characteristics

Hydrogels are three-dimensional networks of hydrophilic monomers and multifunctional linkers that form a flexible and water-laden structure. Hydrogels can undergo swelling and shrinkage suitable for facilitating controlled drug release. Hydrogels can be made from either natural or synthetic monomers. Natural hydrogels, known for their biocompatibility and ability to degrade into non-toxic components, have been extensively studied in tissue engineering. Examples of natural polymers include hyaluronic acid, chitosan, and collagen. However, natural hydrogels often exhibit weaker mechanical strength, challenges in precise formulation and drug loading, and potential immunogenicity [155]. In contrast, synthetic hydrogels are composed of polymers not found in nature. Common synthetic polymers include poly(ethylene glycol) (PEG), poly(vinyl alcohol) (PVA), and poly(hydroxyethyl methacrylate) (PHEMA) [156]. Synthetic polymer hydrogels are highly customizable for specific applications, exhibit good batch-to-batch reproducibility, and have extended stability. However, they may face challenges related to clearance and the formation of toxic by-products during degradation.

Hydrogels achieve cross-linking through physical mechanisms such as chain entanglement, hydrogen bonding, hydrophobic interactions, complexation, or crystallite formation. They can also undergo chemical cross-linking through covalent interactions between the polymer and crosslinker. Furthermore, hydrogels can integrate functionalized components that respond to biological stimuli, making them adaptable to the surrounding environment. This adaptability is significant for in situ forming hydrogels, which crosslink as the temperature increases from room temperature to body temperature, as well as for controlled drug release triggered by factors like pH or photostimulation [157,158,159,160].

#### 4.10.2. Clinical Applications

Hydrogels, owing to their biocompatibility and adjustable drug release profiles, have been studied for drug delivery in various ocular conditions. They can transport drugs to various target locations through different administration methods. Hydrogels are also widely used in combination with different nano-based formulations like NPs, nanomicelles, microneedles, and nanofibers to prolong drug retention and release on the human eye. Swarup et al. used a PNP hydrogel in a mouse model of alkali injury-induced symblephara to prevent fornical shortening and conjunctival fibrosis after injury [161]. Yazdanpanah et al. developed a thermoresponsive hydrogel using decellularized porcine cornea ECM (COMatrix) and demonstrated that this hydrogel improved the attachment and proliferation of human corneal epithelial cells and reduced TNF-α expression in vitro. This suggests its potential use as an ocular surface bandage [162]. Gau et al. explored an anti-VEGF-loaded supramolecular hydrogel that inhibited vascular growth in the retina and reduced CNV by providing extended and controlled release of the anti-VEGF agent. These hydrogels also exhibited the ability to reduce reactive oxygen species and local inflammation. This suggested the potential to replace current anti-VEGF therapy [163]. Cocarta et al. reported a hydrogel implant for transscleral drug delivery to treat retinoblastoma. A bi-layered design with an inner hydrophilic layer bound to a chemotherapeutic agent and the outer hydrophobic layer forms a barrier to prevent cytotoxicity of the delivered chemotherapeutics [164]. Researchers continue to explore and innovate in this field to improve the effectiveness and safety of ocular DDSs using hydrogels.

### 4.11. Microneedles

#### 4.11.1. Main Characteristics

Microneedles (MNs) are small patches with tiny needles under 1 mm in length. They enable localized drug delivery in ocular DDSs, enhancing penetration through ocular barriers for improved therapy. MNs are minimally invasive, making them more acceptable to patients than intravitreal injections [165]. MNs come in various types based on their shape and drug delivery method: (1) Solid MNs create micro-holes to enhance drug permeability. (2) Hollow MNs use pressure to deliver liquid-loaded drugs. (3) Coated MNs have a drug layer that dissolves upon administration. (4) Dissolving MNs release drugs as their matrix dissolves. (5) Hydrogel-forming MNs absorb tissue fluids, promoting drug diffusion [166,167,168].

Although MNs provide minimally invasive and relatively painless methods for direct drug delivery, MNs for ocular DDSs face clinical limitations, including challenges in ensuring precise and consistent insertion and drug delivery, potential for eye damage, and patient acceptance. Regulatory approvals, standardization, and manufacturing costs are also factors to consider. More clinical studies and safety assessments are needed to establish their effectiveness and safety [165,167].

#### 4.11.2. Clinical Applications

Recent progress in pharmaceutical technology has enabled microneedles (MNs) to offer targeted, less invasive, and highly effective drug delivery for various ocular conditions. Over the past decade, a broad spectrum of clinical applications has been explored, spanning from the treatment of keratitis, glaucoma, age-related macular degeneration, uveitis, and retinal vascular occlusion to retinitis pigmentosa [169]. Shi et al. created a dissolving MN patch loaded with fluconazole for treating fungal keratitis in rabbits. The MN patches, made from 30% PLA-HA, penetrated the corneal epithelium without irritating, increased drug residence time in the conjunctival sac by 2.5 h, and provided higher drug bioavailability than conventional eye drops and intrastromal injection [170]. Amer and his colleagues also found that MNs with interlocking features provided an 80% increase in adhesion strength and a slight increase in penetration force compared to microneedles without such features [171]. Matadh and colleagues showed that polymer-coated polymeric (PCP) MNs, a novel approach for controlled drug delivery, released drugs gradually and over a more extended period compared to uncoated MNs, which released the entire drug instantaneously in an ex vivo porcine eye mode [172].

### 4.12. Novel Gene Therapy with Nanotechnology-Based Ocular Delivery Techniques

Gene therapy has emerged as a promising approach to address genetic diseases since the 1960s and is propelled by advances in recombinant DNA technology. In the realm of ophthalmology, gene therapy has gained substantial traction, particularly in targeting retinal diseases, notably those affecting the retinal pigment epithelium [173]. Gene therapy relies on vectors to transport the desired genetic material into host cells. There are two primary categories of vectors: nonviral and viral vectors.

Nonviral delivery systems offer several advantages, including the ability to carry large plasmid DNA, minimal immunogenicity, and a reduced risk of side effects. Nanoparticles (NPs) have been widely used in nonviral gene delivery. These NPs can accommodate sizable plasmid DNA, exhibit safety profiles, sustain long-term protein expression, and pose minimal risk of insertional mutagenesis. Among the most stable NPs are those primarily composed of cationic lipids, PEG lipids, and cholesterol, which are also effective carriers of RNA [174]. Carbon dots (CDs) represent an emerging class of non-viral nanocarriers designed for delivering genes to retinal cells. These CDs possess distinctive physicochemical characteristics, including optical, electronic, and catalytic properties, which render them highly suitable for a wide range of applications, including biosensing, imaging, drug delivery, and, notably, gene delivery [175]. Despite the above advantages of nonviral vectors, challenges related to retinal anatomical barriers and pH sensitivity, which impact the efficiency of gene delivery and the duration of gene expression, are still important issues that need more effort.

Viral vectors use replication-deficient viruses to introduce genetic material into cells, both in vivo and in vitro. Adenoviruses, retroviruses, lentiviruses, and adeno-associated viruses (AAV) are commonly used in gene therapy, with rAAVs being considered a safe and reliable gene delivery method [176]. The CRISPR/Cas system, a revolutionary genome engineering tool, holds promise for treating diverse human diseases, with AAV vectors serving as the primary delivery method for CRISPR applications in the retina [177].

RNA interference (RNAi) therapy has been widely studied for retinal conditions like age-related macular degeneration and glaucoma, which involve specific retinal cell dysfunctions. The effective delivery of therapeutic RNAi to targeted retinal tissues is crucial for success. Lipid-based nanoparticles have shown promise as carriers for RNAi therapeutics, achieving efficient gene silencing in the retinal ganglion cell layer [178]. In summary, gene therapy, RNAi therapy, and CRISPR/Cas technology are emerging as promising strategies for the treatment of various retinal diseases. The development of precise delivery systems to target these gene therapeutics to specific sites within the human eye is of paramount importance.

## 5. Controlled Drug Delivery Systems

The sustained release characteristics in conventional DDSs are mainly based on a diffusion-controlled mechanism which might be affected by the in vivo metabolism environment and lead to probable acute adverse effects. Therefore, controlled drug delivery systems (CDDSs) are gaining attention nowadays for being able to more precisely deliver drugs while minimizing the possible side effects [179]. The majority of CDDSs rely on drug carriers that can react to various types of triggers, whether they are physical or chemical, such as magnetic fields, temperature changes, ultrasound, light, redox reactions, pH shifts, and biological signals like metabolites and enzymes [180]. Among these stimuli, the use of light as an external trigger in CDDSs offers distinct advantages.

Photoactivated nanomaterials for targeted DDSs enables precise focusing, allowing for both spatial and temporal control over the release of therapeutic agents in diseased cells or tissues. Typically, an effective light-triggered DDS involves processes including photolysis, photoisomerization, photo-crosslinking/de-crosslinking, and photoreduction. Both ultraviolet and visible light irradiation can be used as the trigger but the high energy photons also decrease the penetrating ability and might cause high phototoxicity. Therefore, development focuses particularly on low-energy photon irradiation such as near-infrared (NIR), which has a wavelength around 700–900 nm with minimal absorbance to hemoglobin, water, and lipids, allowing for safer and more effective drug delivery [181].

In ocular DDSs, light activation is also an attractive remote triggering method for precise drug release. Various mechanisms for initiating the action of light on liposomes have been devised. The effectiveness of light-activated liposomes, when applied topically, hinges on their capacity to undergo endocytosis within the corneal and conjunctival epithelial cells. In such scenarios, light activation serves as a means to initiate the release of drugs within these cells and facilitate delivery into the cytoplasm [182]. Giannos et al. conducted an in vitro study on photokinetic drug delivery and concluded that pulsed NIR light irradiation can be used to enhance the permeation of Bevacizumab, Ranibizumab, and Aflibercept through human sclera tissue [183]. Wang et al. also developed a multifunctional anti-scarring platform (PVA@rGO-Ag/5-Fu) that combines photothermal capabilities, antibacterial properties, and drug delivery functionality. PVA@rGO-Ag exhibited precise localized photothermal lethality towards both conjunctival fibroblasts and bacteria when exposed to 808 nm NIR radiation, showing the potential of an efficacious anti-scarring strategy for glaucoma surgery [184]. Kari and his colleagues developed a novel drug delivery system by creating a hyaluronic acid–lipid conjugate for light-activated indocyanine green liposomes. This system demonstrated both stability and mobility in vitreous and plasma samples, suggesting its potential applicability for intravenous and intravitreal administration routes [185].

## 6. Conclusions

Nanotechnology-based drug delivery systems are increasingly prominent in clinical pharmacology and biomaterial science, particularly in ophthalmic treatments. These nanocarriers enhance drug permeability, stability, and targeted release, improving drug bioavailability and reducing dosing frequency. For examples, topical cyclosporine A was developed to enhance tear production for patients with dry eye disease. However, the initial cyclosporine A solutions were prepared using oil-based solvents like castor oil or corn oil which led to adverse effects like blurred vision and low bioavailability. Cequa^®^ is an FDA approved aqueous nanomicellar solution composite with cyclosporine A 0.09% that showed better efficacy than the oil-based vehicle group in phase II/III studies [186]. FDA approvals of various ophthalmic formulations with nano-based delivery systems underscore their clinical potential (Table 2). Nevertheless, clinical concerns persist, primarily regarding potential nanomaterial toxicity and degradation byproducts, prompting safety scrutiny. Previous studies have shown that tetrazolium-based assays, including MTT, MTS, and WST-1, are commonly used methods to assess cell viability and cytotoxicity. Biomarkers such as IL-8, IL-6, and the tumor necrosis factor may be measured to evaluate inflammatory or immune responses. A lactate dehydrogenase assay could be used in detecting cell integrity. Diverse cell lines are used in nanocarrier research for in vitro studies while several animal models are used as in vivo models. The toxicity or safety data generated using these different models have been inconsistent and sometimes conflicting. Consequently, it is difficult to quantify and definitively categorize these nanomaterials as more or less toxic to biological systems [187]. The complexity of nanocarrier materials may entail cost challenges, affecting affordability and scalability. There are many different upscaling methods that can sometimes lead to unexpected changes in nanoparticle properties. For instance, in an early study involving the upscaling of nanoparticles using an emulsion method, it was observed that increasing the impeller speed and agitation time resulted in reduced particle size [188]. Furthermore, in ocular DDSs, there is still a limitation on nanocarriers for treatment of posterior segment diseases via noninvasive administration routes. Addressing this challenge requires further innovative research to enhance nanocarrier penetration and achieve targeted delivery. In conclusion, while nanocarriers offer promise for ocular drug delivery, a thorough evaluation, including of safety, therapeutic requirements, market demand, scale-up capabilities, and regulatory aspects, is essential to understand their advantages and limitations.

## Figures and Tables

**Figure 1 ijms-24-15352-f001:**
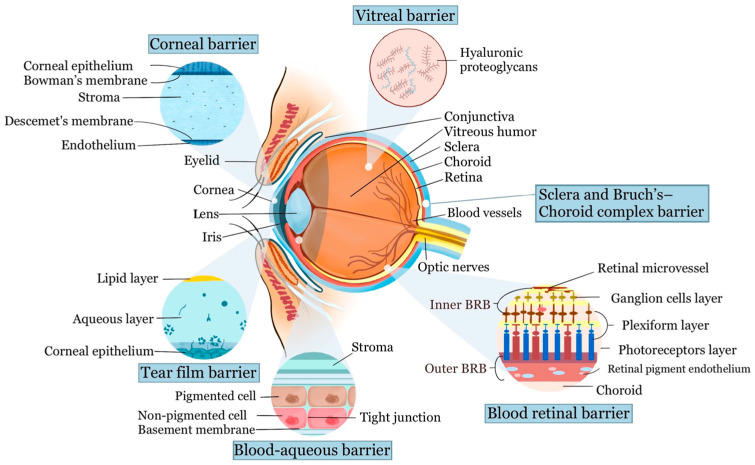
Schematic diagram of the ocular anatomy and the physiological barriers to ocular drug delivery. Reproduced with permission from reference [7]. Copyright 2021, Springer Nature (Creative Commons Attribution 4.0 International License).

**Figure 2 ijms-24-15352-f002:**
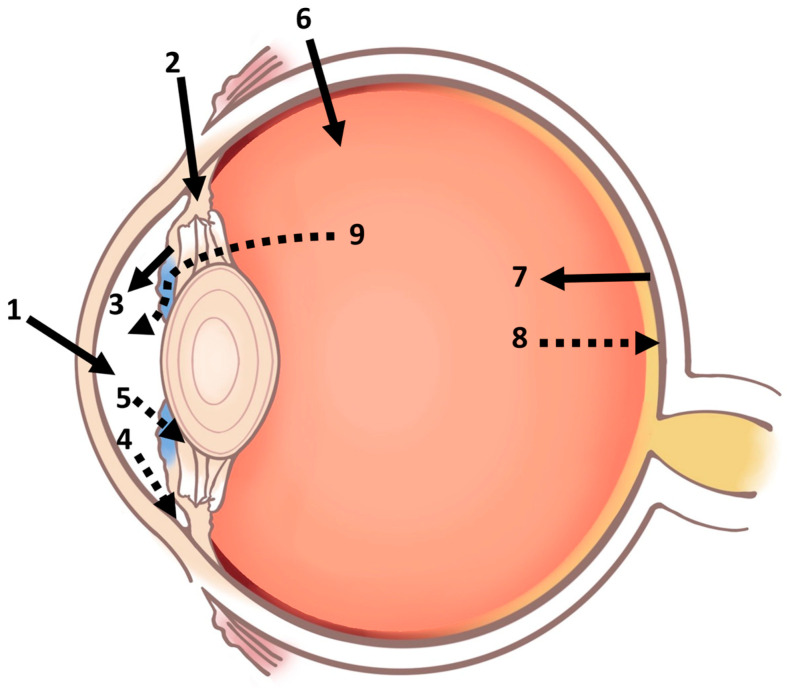
Schematic diagram of the ocular pharmacokinetics. There are multiple pathways for drugs to enter and exit the eye, such as (1) drug absorption via the cornea from tears, (2) drug absorption through the sclera and conjunctiva, (3) drug distribution from the bloodstream to the anterior segment through BAB, and (4) drug elimination via the trabecular meshwork and Schlemm’s canal and (5) from the aqueous humor across the BAB. Moreover, (6) drugs may be directly administered into the vitreous or (7) gain access to the posterior segment of the eye through the BRB. Lastly, drugs can be eliminated from the vitreous via both (8) posterior and (9) anterior routes. (Solid arrows: entering route; Dotted arrows: route of elimination).

**Figure 3 ijms-24-15352-f003:**
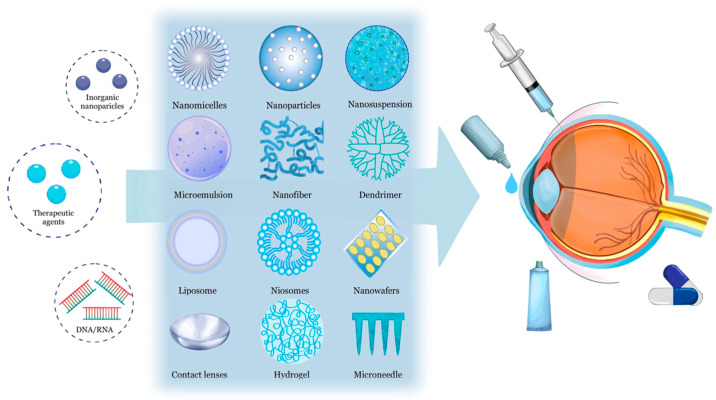
Diagram of nanotechnology-based ocular drug delivery systems. Adapted from Ref. [36]. Copyright 2023, Springer Nature (Creative Commons Attribution 4.0 International License).

**Table 1 ijms-24-15352-t001:** Summary of routes of administration, applications, benefits, and challenges in ocular delivery.

Type	Method	Area	Clinical Applications	Benefits	Challenges
Systemic	Intravenous/Oral	–	Ocular infection, Ocular hypertension, Uveitis, Optic neuritis	High patient compliance	BOB, low bioavailability systemic toxicity caused by high dosing
Topical	–	On the surface of the cornea	Keratitis, uveitis, conjunctivitis, scleritis, episcleritis, blepharitis	High patient compliance, self-administration, non-invasiveness	Tear dilution/turnover tear film and cornea barriers, efflux pumps
Intraocular	Intracameral	Into the anterior chamber	Anesthesia, pupil dilation, endophthalmitis	Direct delivery to the target location, lower dosing, BRB avoidance, higher efficiency	Poor patient compliance, invasiveness, drug toxicity, puncture-related complications (pain, bleeding, vitreous hemorrhage, ocular hypertension, retinal detachment, endophthalmitis, lens and optic nerve damage)
	Intravitreal	Into the vitreal body	AMD, RVO, DME, endophthalmitis, uveitis, CMV retinitis		
	Subretinal	Between neurosensory retina and RPE	AMD, DME, cell therapy for inherited retinal dystrophies [33]		
	Intrastromal	Into the corneal stroma	Keratitis		
	Suprachoroidal	Between the sclera and choroid	Uveitic macular edema and DME [34]		
	Subconjunctival	Beneath conjunctiva	Keratitis, corneal neovascularization [35]		
Periocular	Posterior juxta scleral	Posterior to the supertemporal limbus down to the sclera	Anecortave acetate (Retaane^®^) for AMD, triamcinolone for DME	Selective delivery to both anterior and posterior segments, avoidance of corneal and conjunctival barriers, long duration of action	Poor patient compliance, invasiveness, drug deposition, puncture- related complications (pain, bleeding, infection), risk of globe rupture or scarring, nerve/muscle damage
	Retrobulbar	Intraconal space	Anesthesia		
	Peribulbar	Outside the four rectus muscles and their intramuscular septum	Anesthesia		
	Sub-tenon	Beneath tenon capsule	Chronic uveitis, macular telangiectasia, anesthesia		

**Table 2 ijms-24-15352-t002:** Some examples of FDA-approved products related to nanotechnology-based ocular drug delivery systems.

Product	Nanocarrier Types	Constituents	Indications
Hylo^®^ gel	Hydrogel	Hyaluronate, sorbitol	Dry eye disease
Dextenza^®^	Hydrogel	Dexamethasone	Ocular inflammation, allergic conjunctivitis
ReSure^®^	Hydrogel	Polyethylene glycol	Corneal incisions
Retisert^®^	Intravitreal implant	Fluocinolone acetonide	Uveitis and macular edema
Artelac Rebalance^®^	Liposome	Vitamin B12	Dry eye disease
Lacrisek^®^	Liposome	Vitamin A, E	Dry eye disease
Visudyne^®^	Liposome	Verteporfin	Wet age macular degeneration
Clinitas Hydrate^®^	Liposome	Carbomer 980	Dry eye disease
Cequa^®^	Micelles	Cyclosporine A	Dry eye disease
AzaSite^®^	Micelles	Azithromycin	Dry eye disease, keratitis, eye inflammation
Durezol^®^	Nanoemulsion	Difluprednate	Postoperative ocular inflammation
Restasis^®^	Nanoemulsion	Cyclosporine A	Dry eye disease
Durezol^®^	Nanoemulsion	Difluprednate	Eye infection and pain
Cationorm^®^	Nanoemulsion	Medical device	Dry eye disease
Ikervis^®^	Nanoemulsion	Cyclosporine A	Keratitis
Xelpros^®^	Nanoemulsion	Latanopros	Open-angle glaucoma
Verkazia^®^	Nanoemulsion	Cyclosporine	Vernal keratoconjunctivitis
Cyclokat^®^	Nanoemulsion	Cyclosporine A	Dry eye disease
Systane^®^	Nanoemulsion	Aminomethyl propanol	Relieve dryness of the eye
Trivaris™	Nanoparticles	Triamcinolone acetonide	Uveitis
BromSite^®^	Nanoparticles	Bromfenac	Postoperative inflammation and pain
Besivance^®^	Nanosuspension	Besifloxacin	Ocular bacterial infection
Tobradex ST^®^	Nanosuspension	Tobramycin dexamethasone	Ocular inflammation and bacterial infection
Inveltys^®^	Nanosuspension	Loteprednol etabonate	Postoperative ocular inflammation and pain
Eysuvis^®^	Nanosuspension	Loteprednol etabonate	Dry eye disease
Lacrisert^®^	Ocular inserts	Hydroxypropyl cellulose	Dry eye disease
Biofinity^®^	Soft contact lens	Silicone hydrogel	Correction of ametropia

**Table 3 ijms-24-15352-t003:** Examples of recent exploratory studies on nanomicelles for the treatment of ocular disorders.

Ocular Disorders	Loaded Agents	Micelles	Description	Reference, Year
DED	Cyclosporine A	VitE-TPGS and OPEE	Increase residence time in tear fluid with a t1/2 value four times greater than Ikervis.	[47], 2020
	Cyclosporin A	mPEG–hexPLA	In vivo transcorneal permeability was improved and nanomicelle formulation was significantly efficacious in preventing corneal graft rejection.	[48], 2018
Keratitis	Acyclovir	PVCL-PVA-PEG, Soluplus^®^	Acyclovir-loaded Soluplus micelles showed homogeneous nanometric particle size and slightly negative Z-potential values, facilitating penetration through the cornea and sclera.	[49], 2018
	Voriconazole	PBA-CSVE	PBA-CSVE-loaded voriconazole polymeric micelles proved to have good therapeutic effects, water solubility, biodegradability, low toxicity, and robust mucosal adhesion.	[50], 2022
	Posaconazole	EPC-TPGS	Desirable stability over a month, slow release without an initial burst, and a significantly higher in vitro antifungal activity in comparison with the drug suspension.	[51], 2023
Glaucoma	Nimodipine (NMD)	Rebaudioside A/TPGS	NMD micelles improved the in vivo permeation, intraocular pressure reduction, and miosis.	[52], 2021
	Metipranolol	Pluronic F127 with Chitosan (0.3–0.8%)	Pharmacological response significantly improved upon the incorporation of chitosan.	[53], 2013
Uveitis	Dexamethasone	CSO-VV-SA	Both CSO-VV-SA nanomicelles and HCO-40/OC-40 mixed nanomicelles showed good retention in rabbit tears and equal delivering efficiency.	[54], 2020
	Everolimus	PVCL-PVA-PEG, Soluplus^®^	Everolimus nanomicelles showed significantly higher permeation across goat cornea than everolimus suspension (*p* < 0.001).	[55], 2021
AMD	Artemisinin	PVP K90 and Poloxamer 407	96.0–99.0% artemisinin was released from the nanomicelles within 8 h in vitro. Artemisinin-loaded nanomicelles show superior anti-angiogenic activity compared to artemisinin suspension.	[56], 2021
	Tacrolimus	PEG- HCO-40 and OC-40 (Cequa^®^)	Tacrolimus nanomicellar formulation lowers the pro-inflammatory cytokines and ROS.	[57], 2020
CNV	Aflibercept	PEG, PPG, and PCL copolymer EPC (nEPC)	Aflibercept-loaded nEPCs can penetrate the cornea in ex vivo models and deliver a significant amount of aflibercept to the retina in laser-induced CNV murine models, causing CNV regression.	[58], 2022

DED = dry eye disease; AMD = age-related macular degeneration; CNV = choroidal neovascularization; VitE-TPGS = D-α-Tocopherol polyethylene glycol succinate; OPEE = octylphenoxy poly(ethyleneoxy)ethanol; mPEG–hexPLA = methoxy poly(ethylene glycol)-hexyl substituted poly(lactide); PVCL-PVA-PEG = polyvinyl caprolactam-polyvinyl acetate-polyethylene glycol graft copolymer; PBA-CSVE = phenylboronic acid-coupled chitosan-vitamin E copolymer; EPC-TPGS = egg phosphatidylcholine combined d-a-tocopheryl polyethylene glycol 1000 succinate; CSO-VV-SA = chitosan oligosaccharide-valylvaline-stearic acid; HCO-40 and OC-40 = hydrogenated castor oil-40 and octyxonyl-40; PVP = polyvinylpyrrolidone; ROS = reactive oxygen species; PPG = poly(propylene glycol).

**Table 4 ijms-24-15352-t004:** Examples of recent exploratory studies on nanosuspensions for the treatment of ocular disorders.

Ocular Disorders	Loaded Agents	Stabilizers	Description	Reference, Year
DED	Ciclosporin A	PVA, PVP, HPMC, HPC, HEC	NS was physically and chemically stable for at least two months and caused less irritation to the rabbits’ eyes compared to the commercial product noted by the Schirmer tear test.	[87], 2011
Conjunctivitis	Ketotifen Fumarate	PLGA Eudragit RL100	Both NSs provide a useful dosage form for ocular drug delivery which can enhance the permeability of ketotifen fumarate.	[88], 2016
Keratitis	Voriconazole	Eudragit RS 100	Voriconazole-loaded NS enhances permeability and antifungal activity, effectively inhibiting Candida albicans growth at a lower concentration (2.5 μg/mL, *p* < 0.05) compared to the commercial voriconazole injection.	[89], 2021
Inflammation	Hydrocortisone	PVP, HPMC, Tween 80	NS sustained drug action was maintained up to 9 h compared to 5 h for the drug solution and showed good stability in room-temperature storage.	[90], 2011
	Ketorolac	Eudragit RL-100	NS increases viscosity and avoids drug loss from the precorneal surface and rapid drainage through nasolacrimal areas.	[91], 2019
Glaucoma	Acetazolamide	Anionic polypeptide, poly-γ-glutamic acid (PG), and the glycosaminoglycan, hyaluronic acid	Enhanced saturation solubility, higher reduction of IOP with a longer duration of spray-dried acetazolamide NS, and sustained drug release were confirmed.	[92], 2020

**Table 5 ijms-24-15352-t005:** Examples of recent exploratory studies on nanofibers for the treatment of ocular disorders.

Ocular Disorders	Loaded Agents	Polymers	Description	Reference, Year
Corneal abrasion	Moxifloxacin pirfenidone	PLGA and polyvinylpyrrolidone (PVP)	The antimicrobial activity of moxifloxacin remained effective when encapsulated in the nanofibers, with a sustained release over 24 h. This nanofiber system holds promise for once-daily dosing for the treatment of corneal abrasions	[104], 2021
Keratomycosis	Amphotericin-B	PLGA/Eu-L/Gellan Gum/Pullulan	Amphotericin B complex retained the antifungal activity with sufficient stability against irradiation-sterilization-induced drug degradation and was less toxic to cornea cells in vitro.	[105], 2020
Inflammation	Dexamethasone (DX)	poly(ε-caprolactone) (PCL)	DX PCL nanofibers exhibited ocular biocompatibility and safety by SD-OCT images and histological analysis of neuroretina and choroid in the rodent eye. The nanofiber could provide controlled DX release for 10 days.	[106], 2019
Glaucoma	Timolol maleate (TM) Brimonidine (BR)	Self-assembling peptide ac-(RADA)4-CONH2	A rapid and complete release of both drugs was achieved within 8 h, while a 2.8-fold and 5.4-fold higher corneal permeability was achieved for TM and BR, respectively	[107], 2020
	Brinzolamide	β-cyclodextrin, hydroxypropyl cellulose, and polycaprolactone	The nanofiber provides more precise dosing and permeation through sheep corneas was almost linear in time, achieving therapeutic concentrations in the receptor medium over 6 h.	[108], 2022
	Timolol maleate (TM)	In situ gelling nanofiber films (PVA and Poloxamer 407)	In vivo administration of the ocular films in rabbits induced a faster onset and a sustained IOP-lowering effect up to 24 h.	[109], 2022
AMD	Bevacizumab	PCL and gelatin form the shell of the nanofibers, PVA in the core	Bevacizumab retained its antiangiogenic activity when loaded into the biodegradable core–shell electrospun nanofibers. These nanofibers have the potential to be an alternative treatment option to frequent intravitreal injections of antiangiogenic agents for AMD	[110], 2018
AMD, DR, and glaucoma	Melatonin (MEL)	PVA, PLA	MEL release rates are based on the nature of the polymer. The fast and complete release was observed in PVA-based samples, while the PLA polymer showed slow, controlled MEL release.	[111], 2023

DR = diabetic retinopathy; IOP = intraocular pressure.

**Table 6 ijms-24-15352-t006:** Examples of recent exploratory studies on liposome-based and noisome-based ophthalmic formulations for the treatment of ocular disorders.

Ocular Disorders	Loaded Agents	Nanocarriers	Description	Reference, Year
DED	Azithromycin (AZM)	Liposome	The corneal permeation of AZM-liposome is approximately 2-fold greater than that of the AZM solution, and AZM-liposome significantly improved symptoms of dry eye in rats compared to hyaluronic acid sodium eye drops	[130], 2018
Fungal keratitis	Rapamycin	Liposome	The severity of corneal lesions in the rapamycin-liposome treatment group was reduced	[131], 2019
	Natamycin (NAT)	Niosome	NAT niosomal dispersion exhibited prolonged drug release (40.96–77.49% over 24 h) in vitro. Superiority in treatment of candida keratitis and better results on corneal infiltration and hypopyon level in vivo.	[132], 2019
Ocular infection	Vancomycin	Niosome	In rabbits infected with MRSA, vancomycin niosomal gels showed a 180-fold increase in antibacterial effectiveness compared to untreated animals and a 2.5-fold improvement compared to those treated with the free vancomycin solution.	[133], 2019
GVHD	Tacrolimus	Liposomes containing bile salts	Liposomes containing bile salts increase the corneal transport of tacrolimus to 3–4-fold compared with conventional liposomes.	[134], 2013
Glaucoma	Latanoprost	Niosome	Nonspecific interactions between latanoprost and different niosomal components vary drug encapsulation efficiency. Latanoprost niosomal Pluronic^®^ F127 gel had the best ability to sustain drug release in rabbits’ eyes without toxic and irritant effects and significantly reduced IOP.	[135], 2020
PCME	Triamcinolone acetonide (TA)	Liposome	Patients with refractory PCME under TA-liposome formulation therapy showed a significant improvement in BCVA and central foveal thickness without IOP increase.	[136], 2019
AMD	Berberine hydrochloride chrysophanol	PAMAM G3.0-coated compound liposomes	PAMAM G3.0-coated compound liposomes exhibited good cellular permeability in human corneal epithelial cells and enhanced bio-adhesion to the corneal epithelium in a rabbit model. Liposomes were proven to possess protective effects in human retinal pigment epithelial cells.	[137], 2019
CNV	Triamcinolone acetonide	Chitosan coated liposomes (CCL)	CCL showed a higher encapsulation efficiency with a highly positive surface charge (+41.1 Mv) that increased retention time, sustained release, and penetration via the corneal mucosal barrier to the vitreous body.	[138], 2020
DR	Epalrestat	Cationic niosomes	Niosomal had higher permeation than an unencapsulated drug in the sclera and showed the capability to encapsulate and carry epalrestat through the ocular barrier to treat the diabetic eye.	[139], 2023

MRSA = methicillin-resistant Staphylococcus aureus; GVHD = graft versus host disease; PCME = pseudophakic cystoid macular edema; PAMAM G3.0 = G3 poly(amidoamine) dendrimer.

**Table 7 ijms-24-15352-t007:** Examples of recent exploratory studies on nanotechnology-based contact lenses for the treatment of ocular disorders.

Ocular Disorders	Loaded Agents	Method Used	Description	Reference, Year
Infection	Ciprofloxacin Moxifloxacin	Soaking commercial hydrogel and silicone hydrogel lenses	Release in the vial for both drugs was rapid, reaching a plateau between 15 min and 2 h while under physiological flow conditions; a constant and slow release was observed over 24 h.	[149], 2015
	Natamycin (NA) Fluconazole (FL)	Soaking commercial silicone hydrogel contact lenses	Limited yeast growth.	[150], 2016
Inflammation	Betamethasone	Commercial contact lenses soaked in vitamin E solutions	Vitamin E can be applied as a hydrophobic diffusion barrier for controlling and sustaining BMZ release from silicone-based soft contact lenses.	[151], 2016
	Dexamethasone	Drug-eluting contact lenses	In a rabbit model following photorefractive keratectomy, weekly use of dexamethasone- eluting contact lenses for 4 weeks proved safe and equally effective as applying 0.1% dexamethasone eye drops four times a day over the same duration in preventing corneal haze.	[152], 2021
Glaucoma	Bimatoprost	Graphene oxide-loaded silicone hydrogel contact lenses	Significant improvement in mean residence time and area under the curve with DL-GO-0.2 μg-BMT-100 contact lens was found in the rabbit tear fluid in comparison to the eye drop solution.	[153], 2021
	Timolol Latanoprost	Micelles-laden contact lenses (CLs-M)	Significant improvement of the mean residence time and bioavailability of CLs-M compared with eye drops. The relative pharmacological availability of CLs-M was 9.8 times as high as the eye drops.	[154], 2019

## Data Availability

Not applicable.

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
