# Peer review of "Overview of Recent Advances in Nano-Based Ocular Drug Delivery"

_ijms, 2023, doi:10.3390/ijms242015352_

Round 1

Reviewer 1 Report

Li-Ching Liu and colleagues have presented an intriguing overview of recent advances in nanosystems-based ocular drug delivery. The paper’s flow and the topics covered are engaging for Pharmaceutics readers. However, there are several areas that require improvement before acceptance.

Comments:

  1. Please elaborate on the concepts related to light activation.
  2. Expand on the details of polymersomes and various shape aspects of materials.
  3. In Section 4.10, the authors seem to have conflated implants with hydrogels, which are distinct classes of systems. It is unclear how Ozurdex fits into the nano concept. Please provide justification for the title proposed.
  4. Table 8, titled “FDA-approved commercially available hydrogels for ocular disorders,” requires careful revision.
  5. The conclusions appear too generic and lack new information. Please include your opinions.

I recommend adding specific examples from literature to the following aspects:

  • These nanocarriers improve drug permeability, stability, and targeted release, enhancing drug bioavailability and reducing dosing frequency.
  • FDA approvals for ophthalmic formulations with nano-based delivery systems underscore their clinical potential.
  • However, clinical concerns persist, particularly regarding potential nanomaterial toxicity and degradation byproducts, necessitating safety evaluations.
  • The complexity of nanocarrier materials may pose cost challenges, affecting affordability and scalability.

Moreover, very few papers actually describe quantitative aspects, which is a significant challenge. Animal models and tissue availability also present challenges.

  1. Please summarize the nanosystems covered in a new figure.
  2. I suggest separating exploratory studies and FDA approved products in tables. For instance, approved nanosuspensions seem to be missing. Also, no exploratory studies are presented for Table 4.

The presentation needs to be coherent and uniform.

Reviewer 2 Report

This is an interesting review about recent advances in nanocarrier-based drug delivery systems for ocular administration. The review is generally well written and easy to follow. However, in some cases the difference between different types of nanocarriers is not clear, and the definition od microemulsion is wrong. I recommend publication of this article after minor corrections listed below:

Lines 50-53- would it be possible to provide reference?

Line 230: ‘Nanotechnoledge’- please correct this word

Line 398/399: Microemulsion definition is wrong. It does not contain micron sized droplets as the name suggests. Microemulsion according to IUPAC ‘ Dispersion made of water, oil, and surfactant(s) that is an isotropic and thermodynamically stable system with dispersed domain diameter varying approximately from 1 to 100 nm, usually 10 to 50 nm.’. Please, change the microemulsion definition in the paper and provide references.

Round 2

Reviewer 1 Report

The authors have diligently revised the manuscript. At this stage, I find no further points of critique or suggestions for improvement.